# Millet Fermented by Different Combinations of Yeasts and Lactobacilli: Effects on Phenolic Composition, Starch, Mineral Content and Prebiotic Activity

**DOI:** 10.3390/foods12040748

**Published:** 2023-02-08

**Authors:** Diletta Balli, Lorenzo Cecchi, Giuseppe Pieraccini, Manuel Venturi, Viola Galli, Marta Reggio, Diana Di Gioia, Sandra Furlanetto, Serena Orlandini, Marzia Innocenti, Nadia Mulinacci

**Affiliations:** 1Department of NEUROFARBA and Multidisciplinary Centre of Research on Food Sciences (M.C.R.F.S.-Ce.R.A), University of Florence, Via Ugo Schiff 6, 50019 Florence, Italy; 2Department of Agriculture, Food, Environment and Forestry (DAGRI), University of Florence, 50144 Florence, Italy; 3Mass Spectrometry Center (CISM), University of Florence, Viale G. Pieraccini 6, 50139 Florence, Italy; 4Department of Agricultural and Food Sciences (DISTAL), University of Bologna, Viale Fanin 42, 40127 Bologna, Italy; 5Department of Chemistry “Ugo Schiff”, University of Florence, Via Ugo Schiff 6, 50019 Florence, Italy

**Keywords:** phenols, oligoelements, combined microrganisms, proteolysis, resistant starch, *Bifidobacterium breve*

## Abstract

Millet is the sixth-highest yielding grain in the world and a staple crop for millions of people. Fermentation was applied in this study to improve the nutritional properties of pearl millet. Three microorganism combinations were tested: *Saccharomyces boulardii* (FPM1), *Saccharomyces cerevisiae* plus *Campanilactobacillus paralimentarius* (FPM2) and *Hanseniaspora uvarum* plus *Fructilactobacillus sanfranciscensis* (FPM3). All the fermentation processes led to an increase in minerals. An increase was observed for calcium: 254 ppm in FPM1, 282 ppm in FPM2 and 156 ppm in the unfermented sample. Iron increased in FPM2 and FPM3 (approx. 100 ppm) with respect the unfermented sample (71 ppm). FPM2 and FPM3 resulted in richer total phenols (up to 2.74 mg/g) compared to the unfermented sample (2.24 mg/g). Depending on the microorganisms, it was possible to obtain different oligopeptides with a mass cut off ≤10 kDalton that was not detected in the unfermented sample. FPM2 showed the highest resistant starch content (9.83 g/100 g) and a prebiotic activity on *Bifidobacterium breve* B632, showing a significant growth at 48 h and 72 h compared to glucose (*p* < 0.05). Millet fermented with *Saccharomyces cerevisiae* plus *Campanilactobacillus paralimentarius* can be proposed as a new food with improved nutritional properties to increase the quality of the diet of people who already use millet as a staple food.

## 1. Introduction

Millet, also known as a pseudo-cereal, with its adaptability to arid climates, low water requirement and short production time, is the sixth-highest yielding grain in the world and a staple crop for millions of people living in semi-arid regions. Besides ecological benefits, millet is known for its health and nutraceutical values, which are higher compared to other common cereals (rice, wheat, corn) [1]. The most important kinds of millet are proso millet (*Panicum miliaceum* L.), pearl millet (*Pennisetum glaucum* L.), finger millet (*Eleusine coracana* L.) and foxtail millet (*Setaria italica* L.) [2]. Millet is constituted of proteins (7–11%), carbohydrates (60–70%) and fiber (2–7%), and it is rich in minerals (e.g., calcium, iron, zinc) and bioactive compounds with health-promoting activities [3]. Specific attention to the nutritional quality and cultivation of some minor cereals such as millet can help to reduce malnutrition in some countries that are characterized by an arid climate [4]. Multiple processing techniques have been proposed to facilitate the consumption of millet in daily diets, including decortication, milling, soaking, cooking, roasting, germination and fermentation [5]. As a result of the different processes, many changes can occur in the cereal’s physical, nutritional and functional characteristics [6]. Of all these methods, fermentation is a cost-effective and low-energy process that can be easily applied to improve food preservation, too. It can be spontaneous, by means of intrinsic bacteria, or performed with selected starter cultures. Born as an Asian-African traditional practice [7], today, the consumption of fermented foods is an increasingly widespread trend also in Europe, which has expressed a renewed interest in fermented cereals as healthy foods with improved nutritional, health and sensorial qualities [8,9]. Cereal-based foods fermented by different yeasts have also been suggested as a probiotics delivery vehicle [10,11]. In some African and Asiatic countries, millet is widely fermented to produce different foods, including alcoholic and non-alcoholic beverages, porridges and bakery products [4,12]. Compared to other common cereals, millet has countless advantages from a nutritional point of view, which are amplified during fermentation. An increased amount of dietary fiber attributed to hemicellulose and cellulose degradation was reported in foxtail millet fermented with *Bacillus natto* [13]. Moreover, the improvement of protein and starch digestibility was observed in pearl millet fermented by natural fermentation and in foxtail millet fermented by lactic acid bacteria [6,10]. After fermentation, millet is able to release bioactive peptides smaller than 10 kDa with health benefits, without losing essential amino acids [14]. Furthermore, calcium, iron, phosphorous and zinc availability was demonstrated to be higher in fermented finger millet [4]. An increase in total phenolic content was observed in pearl millet fermented with *Rhizopus azygosporus* [15] and with a mix of yeasts and Lactobacilli [16], as well as in foxtail millet fermented with *Bacillus natto* [13]. To the authors’ knowledge, only one study on finger millet fermented by natural fermentation reported a reduction in total phenolic compounds ranging from 6% to 30% [17].

The purpose of this study is to propose new ingredients based on fermented millet characterized by better nutritional properties that can help to extend the consumption of this minor cereal also in Europe. Because the effects of fermentation are strictly dependent on the metabolic activity of the microorganisms involved, the impact of different microorganism combinations on the content of macronutrients, minerals and bioactive compounds was studied. Three fermentations were performed on Italian pearl millet by two different companies using (i) *Saccharomyces boulardii*, (ii) *Saccharomyces cerevisiae* plus *Companilactobacillus paralimentarius* and (iii) *Hanseniaspora uvarum* plus *Fructilactobacillus sanfranciscensis* and applying consolidated fermentation methods within the companies. Phenolic compounds, resistant starch, slow-digestible starch, total starch, mineral content and oligopeptides were evaluated before and after fermentation. *Bifidobacterium breve* B632 was selected to test and compare the prebiotic activity of unfermented and fermented millet doughs.

## 2. Materials and Methods

### 2.1. Chemicals

All solvents, sodium hydroxide (≥98%), sulfuric acid (95.0–98.0%), ferulic acid, vitexin and vitexin-2-*O*-rhamnoside standards were purchased from Sigma Aldrich (St. Louis, MO, USA). Ultrapure water was obtained by the Milli-Q-system (Millipore SA, Molsheim, France). The *Fructilactobacillus sanfranciscensis* I4 strain belonged to the the culture collection of the Department of Agricultural, Food and Forestry Systems of the University of Florence (Florence, Italy). The other strains used in this study belonged to the collection of the FoodMicroTeam s.r.l. company (Florence, Italy).

### 2.2. Millet Samples and Fermentation Processes

Four different millet samples, belonging to different genera, were purchased. Three of them were purchased from Nigerian local markets: pearl millet (*Pennisetum glaucum* L., code PMN), finger millet (*Eleusine coracana* L., code F_G_M) and foxtail millet (*Setaria italica* L., code F_X_M). The other pearl millet (PMI) was from Insesto s.r.l.farm located in Massa Marittima (Grosseto, Italy), and it was fermented in three different ways by two different companies. Fermentation time and temperature were selected according to the procedures routinely used by the two companies, La BIOTRE and FoodMicroTeam. The first fermentation process was performed by the LaBIOTRE s.r.l company (Florence, Italy) by the addition of 0.2% of *Saccharomyces boulardii* to the unground millet flour after 24 h. Fermentation lasted 72 h, and the liquid medium was dried. A spray dryer was used (B-191; Büchi Milan, Italy) with the liquid medium pneumatically atomized into a vertical co-current drying chamber with inulin as a carrier (50% of final dry weight). The fermented dried sample obtained with this procedure was named FPM1.

The other two fermentation processes were performed by the FoodMicroTeam s.r.l. (Florence, Italy) by mixing the ground millet flour with water (50:50 *w*/*v*) and inoculating two different combinations of yeasts and lactic acid bacteria (LAB): *Saccharomyces cerevisiae* SFY261 + *Campanilactobacillus paralimentarius* Fr L19, obtaining FPM2, and *Hanseniaspora uvarum* SFY309 + *Fructilactobacillus sanfranciscensis* I4, obtaining FPM3. Cultures of each microorganism were centrifuged 5000 rpm for 20 min (Hermle Z 323 K, Hermle Labor Technik, Wehingen, Germany, washed in physiological solution and resuspended to obtain an initial cell density in the pearl millet dough of approximately 1.50 × 10^6^ CFU/g of yeast and 1.50 × 10^7^ CFU/g of LAB. Fermented doughs were dried at 30 °C for 18 h with a Biosec Domus B10 (Tauro essiccatori, Vicenza, Italy) and stored under vacuum before the analyses. At the end of fermentation, microorganism enumeration, pH and total titratable acidity were determined. The pH values were determined by a pH meter (Metrohm Italiana Srl, Varese, Italy) with a food penetration probe. Total titratable acidity (TTA) was measured by an automatic titrator (model FLASH, Steroglass, San Martino in Campo, Perugia, Italy). Ten grams of sample in 90 mL of distilled water were titrated with 0.1 M NaOH to a final pH of 8.5. TTA was expressed as the volume of NaOH used (mL) [18]. The analyzed samples are reported in Table 1.

### 2.3. Microorganism Growth Conditions and Enumeration

The LAB used in this study were *Companilactobacillus paralimentarius* Fr L19, isolated from a spontaneous fermentation of einkorn, and *Fructilactobacillus sanfranciscensis* I4, previously characterized for its antioxidant and anti-inflammatory properties [19]. The selected yeasts were *Saccharomyces cerevisiae* SFY261 and *Hanseniaspora uvarum* SFY309, both of which were isolated from a spontaneous fermentation of vegetables, and *Saccharomyces boulardii*. LAB were routinely propagated for 24 h at 30 °C in MR3i liquid medium before being used as inoculum. The MR3i liquid medium contained (in g/L): maltose 20, glucose 6, fructose 6, polypeptone 10, meat extract 5, yeast extract 12, sodium gluconate 2, sodium acetate trihydrate 5, ammonium citrate trihydrate 2, di-potassium hydrogen phosphate 2, magnesium sulfate heptahydrate 0.2, manganese sulfate tetrahydrate 0.05, cysteine-HCl 0.5, vitamin mix 1 mL, Tween 80 1 mL and fresh yeast extract 15 mL, with a pH of 5.6 [20]. Yeasts were cultured for 24 h at 30 °C in MYPG medium, containing (in g/L): malt extract 5, yeast extract 3, meat extract 5 and glucose 10. Microorganism enumeration was performed as follows: 10 g of the millet dough samples were transferred into 90 mL of a sterile physiological solution and homogenized. LAB and yeasts were diluted, and 100 μL of the suspensions were plated on the appropriate media (MR3i agar medium and MYPG agar for LAB and yeasts, respectively) using the pour plate method. LAB colonies were counted after incubation for 48–72 h at 30 °C under anaerobic conditions, and yeast colonies were counted after incubation for 48 h at 30 °C under aerobic conditions. Plate counts were performed in triplicate.

### 2.4. Phenolic Compound Extraction

Free phenols were extracted from all the unfermented and fermented samples according to the method of Balli et al. [21]. Briefly, twenty mL of acidic MeOH (1% HCl) was added to 2 g of the defatted flour, sonicated for 30 min, stirred with a magnetic stirrer (IKA Labortechnik, Staufen, Germany) for 12 h and centrifuged (5000 rpm, 10 min). The residue was re-suspended in 25 mL of the extractive mixture, sonicated for 30 min (DK Sonic, 42 kHz, Shenzhen, China) and stirred for 2 h. The two collected supernatants were brought to a volume of 50 mL in a flask. The total phenols in the unfermented millet flours were extracted by applying two different hydrolytic procedures according to Balli et al. [21]. The acidic hydrolysis extraction was performed starting from 1 g of defatted flour combined with 25 mL of MeOH/H_2_SO_4_ 1.2 M with the aid of an ultrasonic bath (DK Sonic, 42 kHz), (180 °C for 55 min). The basic hydrolysis was performed starting from 1 g of defatted flour with 40 mL of NaOH 4 M and stirring the solution at room temperature for 4 h. The obtained extracts were then centrifuged (14,000 rpm for 10 min) and analyzed by HPLC-DAD and HPLC-DAD-MS/MS. Only the acid extraction, which was the most effective method in the recovery of the total phenolic compounds, was then applied to the fermented millet samples.

### 2.5. Peptide Extraction

Peptides naturally present in the unfermented sample and those derived from fermentation processes were recovered from PMI, FPM1, FPM2 and FPM3. Briefly, 100 mg of sample was added to 10 mM Tris HCl pH 6.8, vortexed and centrifuged for 10 min at 4 °C (14,000 rpm). Supernatant was then filtered using 10 kDa MWCO filter (Amicon Centrifugal-Merck Millipore, Arklow, Co Wicklow, Ireland) to recover the peptides for the following analytical characterization [22]. 

### 2.6. HPLC-DAD and HPLC-DAD-MS/MS Analyses 

All the extracts were analyzed using a HP 1260L liquid chromatograph equipped with a DAD detector (Agilent Technologies, Palo Alto, CA, USA) and a Raptor column ARC-18 (150 × 3 mm, 5 μm, Restek, Bellefonte, PA, USA). The elution method was previously described by Balli et al. [21]. Briefly, it started with 100% of solvent A (H_2_O at pH 3.2 by HCOOH) and 0% of solvent B (CH_3_CN), with 42 min total analysis time, 0.8 mL/min flow rate and 10 µL injection volume. The UV–vis spectra ranged from 200 nm to 500 nm, and the chromatograms were acquired at 330 and 350 nm. The MS analysis was conducted as previously reported by Balli et al. [21]. The ultrafiltered fermented protein samples were analyzed by nano-liquid chromatography coupled with high-resolution mass spectrometry (nLC-HRMS/MS) via a nanoelectrospray interface, as described in Dani et al. [23]. After concentration in a centrifuge under vacuum, the samples were resuspended in 20 µL of 0.5% acetic acid and injected into the nLC-HRMS/MS instrument (1 µL volume) (Thermo Scientific, Bremen, Germany). After chromatographic separation applying an elution program in a linear gradient (from 2% to 95% acetonitrile: water 0.1% formic acid 80:20, *v*/*v*, versus 0.1% aqueous formic acid), data-dependent acquisition was used, combining 5 MS/MS experiments with one HRMS scan (at 60,000 resolution). A database of millet proteins was created with UniProtKB (taxonomy millet) on 9 February 2022 and used in Mascot (version 2.4, Matrix Science Ltd., London, UK). The database was interrogated using no enzyme and two variable modifications (oxidation of methionine and N-term acetylation), together with a tolerance of 10 ppm for the monoisotopic precursor ion and a 500-millime unit for product ions. A 1% FDR (false discovery rate) was used. Protein identification was accepted on the base of the probability score sorted by Mascot.

### 2.7. Quantitation of Phenolic Compounds by HPLC-DAD

Flavonoids and phenolic acids were quantified according to Balli et al. [21]. In particular, five-point calibration curves (linearity range 0.00–1.23 µg; R^2^ = 1.000) of vitexin and vitexin 2-*O*-rhamnoside (purity ≥ 99%) were used to quantify flavonoids at 350 nm. Phenolic acids were quantified using a five-point calibration curve with ferulic acid as the external standard (purity ≥ 99%) at 330 nm, with a linearity range of 0.00–0.21 µg (R^2^ = 1.000). 

### 2.8. Determination of Resistant, Digestible and Total Starch

Resistant starch (RS), slow-digestible starch (SDS) and total starch (TS) were extracted and determined in all the unfermented and fermented millet samples according to AACC method 32-40.01 [24] using the kit K-RSTAR (Megazyme International Ireland Ltd., Wicklow, Ireland). Measurements were taken in triplicate. Absorbance was measured at 510 nm using an Agilent 8453 G1103A spectrophotometer (Agilent Technologies, Santa Clara, CA, USA).

### 2.9. Mineral Content

Macro- and microelement analysis was performed according to Bellumori et al. [25] on the unfermented (PMI) and fermented (FPM1, FPM2 and FPM3) pearl millet. Briefly, 0.5 mg of dried sample was digested with 10 mL of HNO_3_ (67%) in Teflon reaction vessels: mineralization was performed in a microwave oven (Mars 5, CEM Corp., Matthews, NC, USA) at 200 °C for 15 min using the program of 1600 W, 100% power. Ultra-pure water was added at the end of mineralization to reach a final volume of 25 mL. An inductively coupled argon plasma optical emission spectrometer (ICP–OES iCAP series 7000 Plus Thermo Scientific, Bremen, Germany) was used to determine the concentrations of Al, Cd, Ca, Cu, Cr, K, Mg, Fe, Mo, Mn, Na, Ni, P, Pb, Zn and S. A standard method for the 24 different elements was applied using the QtegraTM Intelligent Scientific Data SolutionTM (ISDS), and the wavelengths selected were 315.8 nm for Ca, 324.7 nm for Cu, 394.4 nm for Al, 228.8 nm for Cd, 283.5 nm for Cr, 259.3 nm for Fe, 769.8 nm for K, 285.2 nm for Mg, 259.3 nm for Mn, 589.5 nm for Na, 231.6 nm for Ni, 178.7 nm for P, 220.3 nm for Pb, 182.0 nm for S and 202.5 nm for Zn quantification. Calibration was performed with several dilutions of the multi-element standard Astasol^®^-Mix (ANALYTIKA^®^, spol. s.r.o., Prague, Czech Republic) in 1% HNO_3_. 

### 2.10. In Vitro Evaluation of the Prebiotic Activity

The prebiotic activity of fermented doughs FPM2 and FPM3 was measured by evaluating the growth stimulation of *Bifidobacterium breve* B632, previously isolated from human feces and deposited into the DSMZ culture collection with the accession number DSM 24706 [26]. The strain was routinely grown from frozen glycerol stock (−80 °C) in tryptone, peptone, yeast extract medium (TPY prepared according to Biavati et al. [27]) for 48 h at 37 °C under anaerobic conditions (Anaerocult A, Merck, Darmstadt, Germany). The growth experiment was performed using modified TPY medium (m-TPY) with halved quantities of substrates supporting growth (tryptone, peptone and yeast extract) and replacing the glucose with FPM2 1% (*w*/*v*) or FPM3 1% (*w*/*v*) as a carbohydrate source. An additional experiment with pearl millet from Italy (PMI) (*w*/*v*) as the sole carbon source was carried out in order to compare the growth with the fermented products. A positive growth control using m-TPY with 0.5% (*w*/*v*) glucose and a negative control in m-TPY with no added carbon source was performed for each condition tested. The experiment with the fermented dough was prepared as follows: 100 mL anerobic glass vessels were filled with 50 mL of fresh m-TPY medium dissolved in water and closed with a ring nut, and the anaerobic atmosphere was created by insufflation of a N_2_/CO_2_ mixture. The medium was then autoclaved at 121 °C for 20 min. A 10% (*w*/*v*) stock solution of fermented dough was prepared by adding 4 g insoluble fermented substrate into 40 mL of m-TPY, which was heated in agitation at 60 °C for 30 min, in order to homogenize the fiber. A 10% (*w*/*v*) glucose stock solution was prepared and filtered. Fermented fiber or glucose was added to each vessel at 1% (*v*/*v*) and 0.5% (*v*/*v*), respectively, to the sterilized medium using a needle. Fresh culture of *Bifidobacterium breve* B632 was subcultured twice in TPY at 37 °C for 48 h, centrifuged, washed and resuspended in m-TPY to achieve an absorbance of 0.7 at 600 nm. Two mL of this suspension (2% *v*/*v*) was used to inoculate each vessel containing the m-TPY medium plus insoluble fermented millet flour (FPM2 or FPM3) or glucose or the negative control with no carbon source. Each condition was tested in triplicate. The vessels were incubated at 37 °C and 130 rpm agitation, and samples were taken from each culture at a pre-established time (0, 24, 48 and 72 h of incubation) for viable bacterial counts, by serial dilution and growth enumeration on TPY agar plate supplemented with mupirocin 100 mg/mL (*v*/*v*). Mupirocin was added to inhibit the growth of lactic acid bacteria [28]. The inoculated plates were incubated at 37 °C under anaerobic conditions for 72 h, and subsequently, the colony count, corresponding to the number of viable cells, was expressed as Log CFU mL.

### 2.11. Statistical Analysis

Each experiment was performed in triplicate, and results were expressed as mean ± SD using EXCEL software (Microsoft, Redmond, WA, USA, version 2019) with in-house routines. Fisher’s LSD (DSAASTAT software v. 1.1, Onofri, Pisa, 2007) was used to identify significant differences between quantitative data. All prebiotic activity assays were performed in triplicate, and the resulting data were expressed as the mean ± SD using EXCEL software (version 2019). The normality of the distributions of prebiotic activity results was evaluated using Shapiro–Wilk’s test, and the homogeneity of variance with Bartlett’s test. Since the samples followed non-normal distributions and non-homogeneity of variance, significant differences between treatments were compared using the Kruskal–Wallis test with post hoc Dunn tests, where *p* < 0.05 was considered statistically significant. Statistical analysis of the results was carried out using R studio. 

## 3. Results

### 3.1. Unfermented Millet Samples: Phenolic and Starch Composition

The chemical composition of four millet varieties, in particular phenolic compounds and resistant, slow-digestible and total starch, were determined in four different millet samples in order to select the sample most suited for the fermentation processes. The phenolic composition of all the millet varieties was investigated by HPLC-DAD-MS, and the identified compounds (Figure 1 and Table 2), belonging to the class of cinnamic acids and flavonoids, were already described in millet [16,21].

The largest amount of free phenols was observed in PMN, pearl millet (*Pennisetum glaucum* L.) from Nigeria: 0.74 mg/g with respect to the average value of 0.30 mg/g found in the other millet varieties. As for the total phenolic content, two different extractive procedures, one using an acidic and one using basic hydrolyses, were compared. The total phenolic amount recovered using the optimized acidic condition (TPA), a method proposed by Balli et al. [21], was approximately 50–70% higher with respect to the phenolic compounds recovered in the basic condition (TPB), with very similar values close to 2.2 mg/g DM for all the analyzed varieties (Table 3A). This result was not completely unexpected and was already reported by Balli et al. [21]. It is worth noting that the basic extraction at room temperature favored polysaccharides’ precipitation entrapping part of the ferulic acid: this mechanical phenomenon led to a loss of ferulic acid of approximately 17–32%, depending on the analyzed sample Consequently, the TPB was not applied to evaluate the phenolic content in the fermented products (Table 3B). As for starch composition, starch from different millet varieties needs to be investigated in order for it to become a better substitute than other conventional starch sources including maize, rice and wheat [29]. Resistant starch (RS) ranged from 9.26 g/100 g in FxM to 24.25 g/100 g in FGM, and the slow-digestible starch (SDS) ranged from 15.41 g/100 g to 38.52 g/100 g (Table 3A). These values were in the same range as those reported by Sandhu et al. [30], higher with respect to those reported by Jayawardana et al. [31] for different finger millet varieties and slightly lower than those reported by Annor et al. [32] and Sharma et al. [33].

### 3.2. Microbiological Composition and Chemical Evaluation of Fermented Millet

PMI was chosen as the millet sample for the fermentation processes, because it contains the highest content of TS and SDS, as well as a total phenolic content (TPA) comparable to all the other samples (Table 3). Furthermore, among the studied samples, PMI was the only variety cultivated from Italy and available in large quantities. The Italian pearl millet was fermented using three different microbial mixtures, obtaining FPM1, FPM2 and FPM3.

Temporally, the first goal of fermentation was aimed at evaluating whether the liquid medium recovered from the fermented millet flour (FPM1), applying a consolidated procedure of the Biotre company, was able to concentrate the bioactive molecules. Because this process required a long time for fermentation and gave a low yield in terms of the dry fermented product, the two other processes were chosen. In particular, two combinations of microrganisms that are already used to ferment fruit and cereals were selected for reducing fermentation time and drying the whole fermented sample. From the first process, FPM1 was obtained, and from the last two tests, FPM2 and FPM3 were obtained. 

Table 4 The initial pH and TTA of the millet doughs were 6.35 ± 0.02 and 1.64 ± 0.01 mL, respectively. Meanwhile, after 24 h of fermentation, all the samples showed a decrease in the pH value and an increase in the TTA. The FPM3 dough was characterized by a lower pH (3.59 ± 0.10) compared to FPM2 (3.93 ± 0.13) and to FPM1 (4.31 ± 0.16), the latter of which showed the highest TTA value (26.10 ± 1.40 mL) with respect to FPM3 and FPM2 (16.94 ± 1.05 mL and 13.76 ± 1.02 mL, respectively). 

As regards the microbial counts, statistically significant differences were detected in the LAB concentrations among the samples, with FPM1 showing the lowest cell densities; regarding yeasts, the *H. uvarum* SFY309 final concentration was lower compared to the two *Saccharomyces* strains. 

The concentration of organic acids (lactic and acetic acids) was higher in the FPM1, reflecting the higher TTA value. Although this sample was not inoculated by LAB, the 72 h fermentation led to the naturally occurring LAB development. Concerning the principal yeast metabolites, the highest concentrations of glycerol and ethanol were found in the FPM3. The amount of total sugars ranged from 1.27 ± 0.31 (g/L) found in the FPM3 to 2.16 ± 0.49 (g/L) of the FPM1.

The chemical composition of the fermented samples was studied, determining: (i) total phenols; (ii) digestible, resistant and total starch; (iii) mineral content. In addition, a preliminary evaluation of the peptide composition of the samples was completed after fermentation, and data were compared with the unfermented millet. 

FPM1 had the poorest result in terms of free and total phenols, because the exclusion of the solid material (remained after the fermentation) determined the loss of part of the bioactive compounds. On the other hand, FPM2 and FPM3, which were obtained by recovering both the solid and liquid components of the fermented flour, showed a higher phenolic content with similar values to that of the unfermented PMI (Table 3B). Concerning the total phenols obtained after acidic hydrolysis (TPA), FPM2 fermented with *Saccharomyces cerevisiae* and *Campanilactobacillus paralimentarius* and FPM3 fermented with *Hanseniaspora uvarum* and *Fructilactobacillus sanfranciscensis* were richer with respect to the unfermented millet sample (Table 3A) with an increase of 22% for FPM2 and 15% for FPM3. These results were in accordance with previous literature data reporting that fermentation increases the percentage of phenolic compounds by different chemical processes [13,15,16]. 

With regard to starch, the percentage of slow-digestible starch increased in FPM1 and FPM3 with a consequent decrease in resistant starch, which was presumably digested by the microorganisms. Significantly, FPM2 presented only a slightly lower content of the resistant starch with respect to the unfermented millet flour, indicating a scarce capacity of the microorganisms to use this substrate for their growth (Table 3A,B). FPM2 and FPM3 had a higher total starch content in comparison to the unfermented millet sample. This result, in accordance with previous data [34], can be explained by the presence of an aliquot of more easily extractable starch in the fermented samples with respect to the unfermented one. 

As for the hydrolyzed proteins, this preliminary investigation was only targeted on millet peptides smaller than 10 kDa produced after the three fermentation processes. According to Table 5, the different microorganisms used for the fermentation processes were able to induce specific protein breakage, thus resulting in the production of different peptides. 

Different groups of peptides were produced starting from the same unfermented cereal. The peptide fragments were attributable to different protein structures, recognized by consulting databases on pearl millet proteins. As for FPM2, the results showed the presence of a large amount of peptides originating from superoxide dismutase with an exponentially modified protein abundance index (emPAI), indicating a relative quantitation of protein in the sample of 233.85, whereas for FPM3, the main peptides originated from Glutathione S-transferase (emPAI, 1.21). It worth noting that these peptides were not detected in the unfermented cereal. This approach did not allow for the identification of already known bioactive peptides of millet, which was previously recognized to exert antimicrobial, antioxidant, antihypertensive, ACE-inhibitory, antiproliferative/anticancer and antidiabetic effects [14,35]. Further efforts are required to investigate the biological properties of these new peptides detected in the FPM2 and FPM3 samples. Moreover, the amount of minerals, particularly calcium and iron (Table 6), increased in all the fermented samples with respect to the unfermented one. This result was in accordance with the previous literature, in which phytase, activated during fermentation, can hydrolyze organic complexes with minerals, making these latter more bioavailable [36]. Furthermore, it was observed that phosphorus also consistently increased after fermentation (from 5360 ppm for PMI to 8194 and 8269 ppm for FPM2 and FPM3, respectively). 

It should be underlined that the existing literature on the evaluation of the chemical changes that occur during the fermentation of millet reports the use of different microorganisms compared to those of our study. Consequently, because the chemical changes occurring during the fermentation processes are microorganism dependent, a strict correlation between our results and those of the few works in the literature on fermented millet cannot be applied.

### 3.3. Evaluation of Bacterial Growth Stimulation on FPM2 and FPM3

*Bifidobacterium* strains, widely present in the gut as part of the normal human microbiota [37], are one of the most studied probiotic genera. They exert beneficial effects on the host, and their proliferation is successfully promoted by various prebiotic substrates [38,39,40,41,42]. In particular, *B. breve* B632 has been chosen for this study, as it exhibits anti-inflammatory activity, the ability to colonize the human gut and protect the integrity of the intestinal epithelium and to stimulate the immune response and compete against pathogens, in addition to having an anti-obesity effect and protective activity against related diseases [26,43,44].

In light of their interesting chemical composition, FPM2 and FPM3 were chosen as substrates to conduct the prebiotic test and were compared with PMI. The capacity of fermented and unfermented doughs to promote the growth of the beneficial bacteria *B. breve* B632 as a carbon source—as compared to glucose and a lack of carbon sources—over a fermentation time of 72 h is reported in Figure 2A–C.

In the presence of FPM2 (Figure 2A) the growth of *B. breve* B632 was highly affected and was sustained for a longer time with respect to the glucose added at the same concentration (G) and compared to the sample without a carbon source (NC). A significant growth increase was in fact observed at 48 h and 72 h compared to glucose (*p* < 0.05), and at 72 h compared to the negative control (*p* < 0.01). Although no significant increase in the viability of *B. breve* B632 was observed between FPM3 and glucose throughout the fermentation time (Figure 2B), FPM3 was able to better support the growth at 72 h in relation to the glucose source. Growth on the FPM3 sample was 1.0 and 0.6 Log CFU/mL lower at 24 and 48 h, respectively, than that on glucose. A significantly higher growth rate was shown in the presence of glucose than without a carbon source at 24 (*p* < 0.05). Figure 2C shows that *B. breve* B632 had a high proliferation rate in the presence of PMI, similar to that reached with glucose, even if it resulted in no significance (*p* > 0.05); as expected, glucose showed greater and significant growth than the carbonless sample (*p* < 0.05) at 24 h. Once again, the maintenance of growth was observed up to 72 h with PMI compared to glucose, although no difference was observed with respect to the NC sample. The prebiotic test carried out with *B. breve* B632 showed that the fermented millet FPM2 was a better substrate for the growth of the microorganism over time, being more effective than the unfermented millet sample.

## 4. Conclusions

Fermentation has been studied in this manuscript as a simple and low-cost technique for preparing new millet-based ingredients with improved nutritional properties. In particular, one mixture of *lactobacilli* and yeast, among the three tested in the study, was capable of conferring an additional nutraceutical and nutritional value compared to the unfermented cereal. The possibility of preparing ingredients/foods with greater nutritional properties by a simple and low-cost method can improve not only the quality of food for people who already use millet as a staple, but it can also favor the greater use of millet in countries that are not accustomed to the consumption of this minor cereal. Furthermore, actions targeted to facilitating/improving the use of millet, which is drought tolerant and known for its agronomic advantages, may be an interesting strategy for coping with climate changes. From a future perspective, this work paves the way for new experiments for identifying a more suitable mixture of microorganisms to obtain millet-based foods with added nutritional value.

## Figures and Tables

**Figure 1 foods-12-00748-f001:**
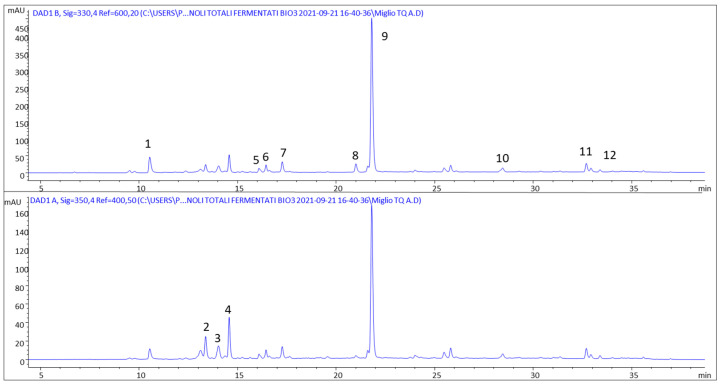
HPLC-DAD phenolic profile at 330 nm (upper trace) and 350 nm (lower trace) of unfermented pearl millet. 1: *N*^1^,*N*^4^-dicaffeoylspermidine; 2: luteolin-(7-*O*-glucopyranosyl)-8-*C*-glucopyranoside; 3: vitexin-2″-*O*-rhamnoside; 4: vitexin; 5: ferulic acid rhamnoside; 6: ferulic acid rhamnoside isomer; 7: isoferulic acid; 8: methyl hydroxycinnamate; 9: methyl ferulate; 10–12: cinnamic acid derivatives.

**Figure 2 foods-12-00748-f002:**
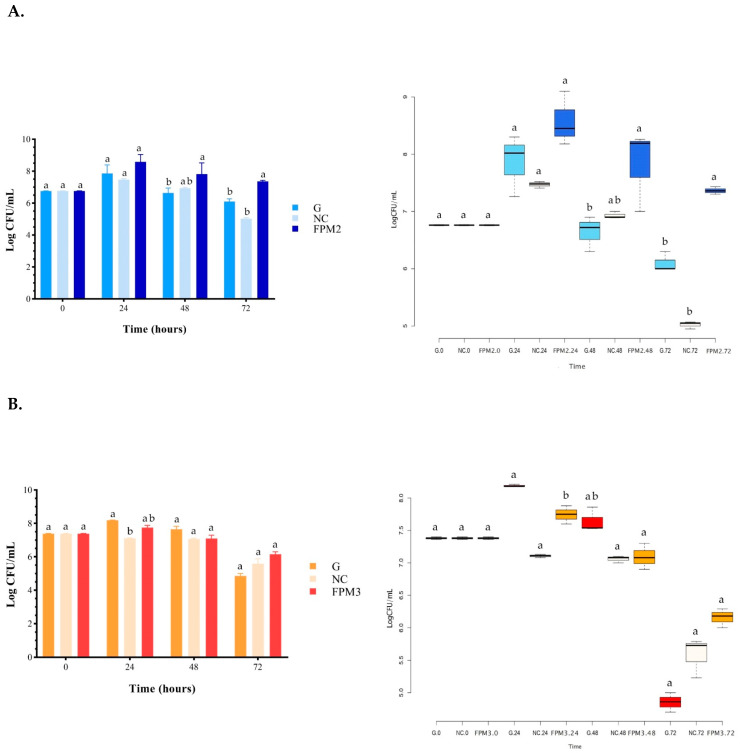
Prebiotic activity evaluation of (**A**) FPM2 (1% in m-TPY), (**B**) FPM3 (1% in m-TPY) and (**C**) PMI (1% in m-TPY) on *B. breve* B632 compared to G (growth on m-TPY with 0.5% glucose) and NC (growth on m-TPY with no added carbon source). CFU, colony-forming units. The values are means of viable cell counts of three independent experiments (±standard deviation). Different letters (a, b) indicate significant differences (*p* < 0.05) according to Dunn’s test.

**Table 1 foods-12-00748-t001:** Analyzed samples: (A) unfermented millet samples; (B) fermented pearl millet from the PMI sample.

A.					
Code	Plant Name	Origin	Color	Plant Genera	Species
PMN	Pearl millet	Nigeria	yellow/brown	*Pennisetum*	*glaucum*
F_G_M	Finger millet	Nigeria	red/brown	*Eleusine*	*coracana*
F_X_M	Foxtail millet	Nigeria	yellow	*Setaria*	*italica*
PMI	Pearl millet	Italy	yellow/brown	*Pennisetum*	*glaucum*
**B.**					
**Code**	**Fermentative Microorganisms**		**Fermentation Time**	**Temperature**	**Drying Method**
FPM1	*Saccharomyces boulardii*		72 h	30 °C	Spray drying
FPM2	*Saccharomyces cerevisiae* + *Companilactobacillus paralimentarius*		24 h	28 °C	Oven drying
FPM3	*Hanseniaspora uvarum* + *Fructilactobacillus sanfranciscens*		24 h	28 °C	Oven drying

**Table 2 foods-12-00748-t002:** Identified compounds in millet samples.

Analytes	[M − H]^−^	Identified Compounds
1	468	*N*^1^,*N*^4^-dicaffeoylspermidine
2	609	luteolin-(7-*O*-glucopyranosyl)-8-*C*-glucopyranoside
3	577	vitexin-2″-*O*-rhamnoside
4	431	vitexin
5	339	ferulic acid rhamnoside
6	339	ferulic acid rhamnoside isomer
7	193	isoferulic acid
8	177	methyl hydroxycinnamate
9	207	methyl ferulate

**Table 3 foods-12-00748-t003:** Phenolic and starch composition of (A) unfermented and (B) fermented samples from only PMI (TPB was not evaluated due to the lower extraction efficiency shown for the unfermented samples in Table 3A). Data are expressed in mg/g and g/100 g ± standard deviation on dry weight as a mean of triplicate. PMN: pearl millet Nigeria; FxM: foxtail millet; FgM: Finger millet; PMI: pearl millet Italy; FPM1: fermented pearl millet with *Saccharomyces boulardi;* FPM2: fermented pearl millet with *Saccharomyces cerevisiae + Companilactobacillus paralimentarius;* FPM3: fermented pearl millet with *Hanseniaspora uvarum* + *Fructilactobacillus sanfranciscensis*.

(A)						
	Phenolic Composition			Starch Content		
	FP mg/g	TPB mg/g	TPA mg/g	RS g/100 g	SDS g/100 g	TS g/100 g
PMN	0.74 ± 0.02 c	1.13 ± 0.05 d	2.26 ± 0.03 b	11.23 ± 1.72 b	23.85 ± 2.15 b	31.34 ± 3.09 b
FxM	0.26 ± 0.04 a	0.52 ± 0.07 a	2.15 ± 0.07 a	9.26 ± 1.12 a	15.41 ± 1.02 a	24.68 ± 2.11 a
FgM	0.30 ± 0.03 b	0.81 ± 0.05 b	2.32 ± 0.06 c	24.25 ± 0.76 c	25.39 ± 1.25 b	49.65 ± 2.68 c
PMI	0.30 ± 0.01 b	1.02 ± 0.07 c	2.24 ± 0.05 b	11.83 ± 0.65 b	38.52 ± 0.48 c	50.35 ± 1.61 c
**(B)**						
	**Phenolic Composition**			**Starch Content**		
	**FP** **mg/g**		**TPA** **mg/g**	**RS** **g/100 g**	**SDS** **g/100 g**	**TS** **g/100 g**
FPM1	0.12 ± 0.01 a		1.20 ± 0.02 a	0.71 ± 0.03 a	3.91 ± 0.78 a	4.62 ± 3.82 a
FPM2	0.27 ± 0.01 c		2.74 ± 0.10 c	9.83 ± 0.24 c	50.18 ± 0.69 b	60.00 ± 0.93 b
FPM3	0.25 ± 0.01 b		2.57 ± 0.03 b	4.03 ± 0.40 b	55.39 ± 1.40 c	59.42 ± 1.79 b

FP: free phenols; TPB: total phenols by basic hydrolysis; TPA: total phenols by acidic hydrolysis; RS: resistant starch; SDS: slow-digestible starch; TS: total starch. a, b, c, d: means within a column with different letters are significantly different (*p* < 0.05).

**Table 4 foods-12-00748-t004:** Parameters compared for the fermented millet samples: pH, total titratable acidity (TTA) expressed in mL, lactic acid bacteria (LAB) and yeast concentration expressed in CFU/g, content of total sugars (maltose, fructose and glucose) and microbial metabolites expressed in g/L or %. Results are expressed as mean ± standard deviation.

**Sample**	**Final pH**	**TTA**	**LAB**	**Yeasts**	**Total Sugars**	**Lactic Acid**	**Acetic Acid**	**Glycerin**	**Ethanol**
		**(mL)**	**(CFU/g)**	**(CFU/g)**	**(g/L)**	**(g/L)**	**(g/L)**	**(g/L)**	**(%)**
FPM1	4.31 ± 0.16 c	26.10 ± 1.40 c	(1.20 ± 0.32) × 10^8^ a	(3.12 ± 0.51) × 10^8^ b	2.16 ± 0.49 b	1.97 ± 0.23 c	0.26 ± 0.02 b	0.36 ± 0.14 ab	0.17 ± 0.06 a
FPM2	3.93 ± 0.13 b	13.76 ± 1.02 a	(3.70 ± 0.42) × 10^9^ c	(5.34 ± 1.50) × 10^8^ c	1.54 ± 0.33 ab	0.23 ± 0.03 a	0.14 ± 0.02 a	0.17 ± 0.02 a	0.09 ± 0.01 a
FPM3	3.59 ± 0.10 a	16.94 ± 1.05 b	(1.95 ± 0.18) × 10^9^ b	(4.25 ± 0.35) × 10^7^ a	1.27 ± 0.31 a	0.49 ± 0.09 b	0.22 ± 0.06 b	0.46 ± 0.10 b	0.36 ± 0.09 b

a, b, c: means within a column with different letters are significantly different (*p* < 0.05).

**Table 5 foods-12-00748-t005:** Oligopeptides (cut off ≤10 KDalton) detected in FPM1 fermented with Saccharomyces boulardi, FPM2 fermented with Saccharomyces cerevisiae + Companilactobacillus paralimentarius and FPM3 fermented with *Hanseniospora uvarum* + *Fructilactobacillus sanfranciscens*. Exponentially Modified Protein Abundance Index (emPAI) indicates a relative quantitation of protein in the sample.

Mass	emPAI	Description	Identification	Fermented Samples		
168,430	0.02	A0A1D8KW99_CENAM	DNA-directed RNA polymera se subunit beta		FPM2	
113,891	0.03	A0A0B5ACT4_CENAM	NBS-LRR-like protein	FPM1	FPM2	FPM3
95,958	0.03	A0A4Y1NYR9_CENAM	Calmodulin-binding transcription activator 4			FPM3
82,491	0.04	A0A024BLE7_CENAM	Photosystem I P700 chlorophyll a apoprotein A2			FPM3
80,220	0.04	E5FQ64_CENAM	Heat-shock protein 90			FPM3
72,966	0.04	A4ZYQ0_CENAM	Chloroplast heat-shock protein 70	FPM1	FPM2	FPM3
69,627	0.05	A0A2R3STY0_CENAM	Putative kinase-like protein TMKL1			FPM3
56,674	0.06	B5TSR3_CENAM	DELLA protein		FPM2	
53,877	0.06	A0A024BLC0_CENAM	ATP synthase subunit beta			FPM3
53,466	0.06	A0A068EUE1_CENAM	Glutathione reductase			FPM3
52,531	0.06	A0A1B0RMG0_CENAM	Purple acid phospatase		FPM2	
48,657	0.22	A0A172DYZ9_CENAM	Calreticulin		FPM2	FPM3
47,705	0.07	M1PSE1_CENAM	Ribulose bisphosphate carboxylase large chain		FPM2	
46,281	0.07	Q8LKI5_CENAM	Opaque-2-like protein	FPM1		
45,994	0.07	B5AKW1_CENAM	eIF-4A	FPM1		
43,045	0.16	I3RJV4_CENAM	DELLA protein	FPM1		FPM3
42,984	0.08	I3RJV7_CENAM	DELLA protein	FPM1	FPM2	
42,983	0.16	I3RJW9_CENAM	DELLA protein		FPM2	
42,870	0.08	A0A076Q103_CENAM	Calcium-dependent protein kinase			FPM3
40,915	0.17	Q94IL8_CENAM	Alcohol dehydrogenase		FPM2	FPM3
39,548	0.08	A0A024BKG2_CENAM	Silicon transport protein		FPM2	FPM3
39,210	0.08	A0A0S1MNE3_CENAM	CaFPM1eoyl CoA O-methyltransferase	FPM1		
38,910	0.08	A0A024BKF6_CENAM	Photosystem II protein D1	FPM1		
36,942	0.09	A0A089N0T7_CENAM	Tubulin beta chain		FPM2	
36,874	0.19	Q8LRN0_CENAM	Glyoxalase II		FPM2	FPM3
35,920	0.09	A0A4Y1NY14_CENAM	Dehydration-responsive element-binding protein 2 A			FPM3
35,714	0.09	W5QKC5_CENAM	Polygalacturonase inhibitor protein 1	FPM1		
32,918	0.1	Q5NKR7_CENAM	Uncharacterized protein 311G2.2			FPM3
31,501	0.1	A0A823A7Z2_CENAM	Aquaporin noduline-26-like intrinsic protein 4-1	FPM1		
31,405	0.11	A0A823A730_CENAM	Aquaporin noduline-26-like intrinsic protein 4-1		FPM2	
30,780	0.11	A0A089MYF7_CENAM	Actin-7-like protein		FPM2	FPM3
30,562	0.11	A0A024BKN6_CENAM	Ribosomal protein L2		FPM2	
30,326	0.11	A0A823A8Q5_CENAM	Aquaporin plasma membrane intrinsic protein 2-1	FPM1		
30,061	0.11	Q8HNK2_CENAM	Cytochrome c oxidase subunit 3			FPM3
29,498	0.11	A0A823ADZ2_CENAM	Aquaporin noduline-26-like intrinsic protein 1-1	FPM1		
29,134	0.24	A6N4D4_CENAM	27 kDA pennisetin		FPM2	FPM3
27,452	0.12	A4ZYP9_CENAM	L-ascorbate peroxidase	FPM1		
27,387	0.12	Q5MJ19_CENAM	RING zinc-finger protein	FPM1		
27,247	0.12	G9M131_CENAM	Phytochrome B	FPM1		FPM3
26,887	0.12	CAPSD_MSVSE	Capsid protein			FPM3
26,336	0.13	A0A7T8J1V5_CENAM	PWWP domain family-like protein			FPM3
25,718	0.13	A0A1B1SJY6_CENAM	TIFPM21	FPM1		
25,675	0.13	G9M0H0_CENAM	GIGANTEA		FPM2	FPM3
25,440	0.13	G9M1I7_CENAM	Phytochrome C		FPM2	
25,133	0.13	A0A822ZYW4_CENAM	Aquaporin noduline-26 intrinsic protein 3-4			FPM3
24,652	0.14	B3SU24_CENAM	Elongation factor 1 subunit alpha		FPM2	FPM3
24,520	0.14	Q6R2L1_CENAM	Photosystem I A apoprotein		FPM2	
22,459	0.15	A6N4D3_CENAM	21 kDa pennisetin	FPM1		
20,863	0.16	Q9M6M2_CENAM	NB-ARC domain-containing protein		FPM2	
18,632	0.18	Q06HR0_CENAM	ATP-dependent Clp protease ATP-binding subunit ClpX1	FPM1		
16,414	0.21	W5QKC6_CENAM	Polygalacturonase inhibitor protein 2			FPM3
15,149	233.85	A4ZYP8_CENAM	Superoxide dismutase		FPM2	
13,526	0.25	A0A024BLF0_CENAM	Ribosomal protein L14			FPM3
12,402	0.62	D8V069_CENAM	Chitinase		FPM2	FPM3
11,282	1.21	A0A0K1DBU0_CENAM	Glutathione S-transferase			FPM3
10,899	0.32	MP_MSVK	Movement protein	FPM1		
10,720	0.32	D7F3V4_CENAM	30S ribosomial protein S19, chloroplastic		FPM2	
9186	0.38	Q32ZI6_CENAM	Truncated vacuolar ATPase subunit C isoform	FPM1		FPM3
5058	0.71	G9M2M4_CENAM	Uncharacterized protein			FPM3

**Table 6 foods-12-00748-t006:** Mineral content in pearl millet (PMI), millet fermented with *Saccharomyces boulardi* (FPM1), millet fermented with *Saccharomyces cerevisiae + Companilactobacillus paralimentarius* (FPM2) and millet fermented with *Hanseniaspora uvarum* + *Fructilactobacillus sanfranciscensis* (FPM3). Data are expressed in ppm, as mean of triplicate ± standard deviation.

Samples	Al	Ca	Cu	Fe	K	Mg	Mn	Mo	Na	Ni	S	Zn
PMI	7.64 ± 0.49 a	156.64 ± 5.82 a	4.75 ± 0.15 b	71.71 ± 0.74 a	2313.08 ± 6.25 c	996.37 ± 18.47 a	14.16 ± 0.24 a	0.46 ± 0.03 b	18.85 ± 3.04 a	2.12 ± 0.03 a	1051.43 ± 20.60 b	37.22 ± 0.29 a
FPM1	13.20 ± 0.64 a	254.12 ± 2.54 b	4.28 ± 0.17 a	71.54 ± 1.13 a	2477.26 ± 14.78 d	1395.48 ± 13.39 d	18.48 ± 0.18 c	0.22 ± 0.08 a	215.42 ± 4.67 d	3.54 ± 0.08 d	733.64 ± 3.31 a	40.23 ± 0.88 b
FPM2	20.47 ± 3.52 b	282.06 ± 5.45 d	6.67 ± 0.12 d	110.41 ± 2.76 c	2093.95 ± 61.92 b	1138.98 ± 4.99 c	18.34 ± 0.32 c	0.30 ± 0.02 a	63.47 ± 3.72 c	3.25 ± 0.04 c	1405.09 ± 30.76 d	56.24 ± 1.99 d
FPM3	21.02 ± 6.26 b	263.26 ± 3.19 c	5.99 ± 0.14 c	96.59 ± 1.01 b	1988.66 ± 12.90 a	1073.04 ± 14.45 b	17.39 ± 0.10 b	0.28 ± 0.07 a	56.65 ± 0.12 b	2.98 ± 0.04 b	1249.59 ± 13.07 c	51.53 ± 0.95 c

a, b, c, d: means within a column with different letters are significantly different (*p* < 0.05).

## Data Availability

All related data and methods are presented in this paper. Additional inquiries should be addressed to the corresponding author.

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
