# Peer review of "Millet Fermented by Different Combinations of Yeasts and Lactobacilli: Effects on Phenolic Composition, Starch, Mineral Content and Prebiotic Activity"

_foods, 2023, doi:10.3390/foods12040748_

Round 1
Reviewer 1 Report
Dear Editors and authors,
1-The abstract of the manuscript needs to be supported by adding some numbers of the results.
2-The names of many bacteria should be written in italics throughout the manuscript. See line 27 and key words
3-The introduction needs some scientific references about the use of Probiotic yeasts in food, I suggest you
Staniszewski, A., & Kordowska-Wiater, M. (2021). Probiotic and potentially probiotic yeasts—characteristics and food application. Foods, 10(6), 1306.
4-Microbial tests, the use of yeasts and bacteria and their activation methods are not clear.
5- The culture medium used to activate lactic acid bacteria is incorrect, see line 132 and 137.
6-Total titratable acidity method (line 117) need to add new reference see (Niamah, A. K., Al-Manhel, A. J., & Al-Sahlany, S. T. G. (2018). EFFECT MICROENCAPSULATION OF Saccharomyces boulardii ON VIABILITY OF YEAST IN VITRO AND ICE CREAM. Carpathian Journal of Food Science & Technology, 10(3): 100-108.
7-Some methods need to add new references, Like Protein extraction , HPLC-DAD e HPLC-DAD-MS/MS analyses , Quantitation of phenolic compounds by HPLC-DAD .and Mineral content.
8-Please add reference of TPY agar plate supplemented with Mupirocin 100 μg/mL, Line 254.
9-Correct the line under Table 3.
10-Conclusions Many of the results obtained during the study contain. Such conclusions cannot be written, they must be rewritten and the results removed from them.
Reviewer 2 Report
The abstract should be more informative by giving real results rather than elastic sentences. Important and main contents should be given. Support the results with some quantitative data. Moreover, no conclusions are provided.
Please choose keywords in such a way that they are not mentioned in the title in addition to helping to understand the concept of the research.
To complete the introduction of the article in the sections related to starch as well as antioxidant and phenolic properties, it is better to use more recent articles. For example, you can use the article 10.1016/j.ijbiomac.2022.11.044 ;
In the entire text of the article, write the unit of time in the SI system as h and min. It is given in the form of hours and minutes in the sections that need to be corrected.
Write the name of "MR3i agar" culture medium correctly.
In line 142 it is said that Plate counts were performed in duplicate, while in section 2.11 it is said that it was done in triplicate. Please make this more clear.
The data relating to "TPB: Total phenols by basic hydrolysis" are not given in Table 2. Please put a column related to these data in that table.
The standard deviation for the results mentioned in all tables and graphs must be given and "significant letters" must be mentioned in all of them.
The analyzes presented in all the results and discussion sections are not sufficient and do not explain the reasons for the observed results. Please review the Results and Discussion section in its entirety. Please compare with similar works after presenting each result. Improve this section carefully.
Font sizes are different in different parts (for example lines 332-334). Please correct them.
Figure 1S and Table 1S should be given in the article.
Conclusion: what is the future of your findings? Conclusion is not insightful, what are suggestions?
Reviewer 3 Report
The topic of the study is of extreme importance in order to cope with the need for new ingredients. Millets do have great potential for that, however, the article needs to be improved. For instance, the author says in the introduction that millets can solve the problem of hunger and malnutrition worldwide which is not correct. It is one of the solutions and not the overall as stated. In addition, it is a region-specific solution as this cannot be used in other regions that also suffer from malnutrition. I invite the authors to revise the introduction and make clear what is the ultimate aim of the article.
The methodology section needs to be clarified. Fermentation was carried out at different places by different groups? Why were some samples spray-dried and others oven-dried? Moreover, what was the basis used to decide fermentation time and temperature?
Throughout the manuscript, chemical formulas need to be in subscripts accordingly.
Conclusion needs to be considerably improved as it is a huge summary of the results and not a conclusion.
Other minor comments:
Line 34: What is meant by flexibility towards climate changes?
Line 43: reprase , as this is to generalist and idealist as the problem of hunger and malnutrition, cannot be solved by an overall solution as authors are claiming. Millets will not solve the world’s problem, only help it.
Line 48: is fermentation really a traditional/household method for millets?
Line 102: Clarify what is LaBIOTRE
Line 104: English check
Line 105: Provide the spray drying conditions
Line 106: Clarify what is FoodMicroTeam
Table 1 and 3: Improve quality (e.g., a and b), place it appropriately
Section 205: How are the extraction conditions define? From other literature or by research group preliminary experiments?
Table 2: Carry out comparative tests
Round 2
Reviewer 1 Report
Dear editors and author,
The authors made all required revisions to improve the manuscript, and I now propose that it be published in its current form.
Reviewer 2 Report
Thanks to respected authors.
The revised manuscript is acceptable in its present form.